# Allergenic Activity of Individual Cat Allergen Molecules

**DOI:** 10.3390/ijms242316729

**Published:** 2023-11-24

**Authors:** Daria Trifonova, Mirela Curin, Ksenja Riabova, Antonina Karsonova, Walter Keller, Hans Grönlund, Ulrika Käck, Jon R. Konradsen, Marianne van Hage, Alexander Karaulov, Rudolf Valenta

**Affiliations:** 1Division of Immunopathology, Department of Pathophysiology and Allergy Research, Center for Pathophysiology, Infectiology and Immunology, Medical University of Vienna, 1090 Vienna, Austria; daria.trifonova@meduniwien.ac.at (D.T.);; 2Laboratory of Immunopathology, Department of Clinical Immunology and Allergy, Sechenov First Moscow State Medical University, 119991 Moscow, Russiadrkaraulov@mail.ru (A.K.); 3Institute of Molecular Biosciences, BioTechMed Graz, University of Graz, 8010 Graz, Austria; walter.keller@uni-graz.at; 4Therapeutic Immune Design Unit, Department of Clinical Neuroscience, Karolinska Institutet, 17177 Stockholm, Sweden; hans.gronlund@ki.se; 5Department of Clinical Science and Education, Södersjukhuset, Karolinska Institutet, 11883 Stockholm, Sweden; ulrika.kack@regionstockholm.se; 6Pediatric Allergy and Pulmonology, Astrid Lindgren Children’s Hospital, Karolinska University Hospital, 17164 Stockholm, Sweden; 7Department of Women’s and Children’s Health, Karolinska Institutet, 17177 Stockholm, Sweden; 8Division of Immunology and Allergy, Department of Medicine Solna, Karolinska Institutet and University Hospital, 17177 Stockholm, Sweden; marianne.van.hage@ki.se; 9Karl Landsteiner University for Healthcare Sciences, 3500 Krems, Austria

**Keywords:** allergy, cat allergy, cat allergen molecule, IgE reactivity, allergenic activity, basophil activation test

## Abstract

More than 10% of the world’s population suffers from an immunoglobulin E (IgE)-mediated allergy to cats which is accompanied mainly by respiratory symptoms such as rhinitis and asthma. Several cat allergen molecules have been identified, but their allergenic activity has not been investigated in depth. Purified cat allergen molecules (Fel d 1, Fel d 2, Fel d 3, Fel d 4, Fel d 6, Fel d 7 and Fel d 8) were characterized via mass spectrometry and circular dichroism spectroscopy regarding their molecular mass and fold, respectively. Cat-allergen-specific IgE levels were quantified via ImmunoCAP measurements in IgE-sensitized subjects with (n = 37) and without (n = 20) respiratory symptoms related to cat exposure. The allergenic activity of the cat allergens was investigated by loading patients’ IgE onto rat basophils expressing the human FcεRI receptor and studying the ability of different allergen concentrations to induce β-hexosaminidase release. Purified and folded cat allergens with correct masses were obtained. Cat-allergen-specific IgE levels were much higher in patients with a respiratory allergy than in patients without a respiratory allergy. Fel d 1, Fel d 2, Fel d 4 and Fel d 7 bound the highest levels of specific IgE and already-induced basophil degranulation at hundred-fold-lower concentrations than the other allergens. Fel d 1, Fel d 4 and Fel d 7 were recognized by more than 65% of patients with a respiratory allergy, whereas Fel d 2 was recognized by only 30%. Therefore, in addition to the major cat allergen Fel d 1, Fel d 4 and Fel d 7 should also be considered to be important allergens for the diagnosis and specific immunotherapy of cat allergy.

## 1. Introduction

Cat ownership is very popular worldwide. In some countries, more than 50% of all households have a cat [1]. Since cats have become a relevant part of our indoor environment, it is not surprising that cat dander is among the most important sources of indoor allergens in Europe and Asia [2,3,4,5,6,7]. The percentage of cat-sensitized patients seems to be particularly high in Northern Europe, Scandinavia and Russia, which may be due to living habits and climate conditions [8,9,10]. A relationship between sensitization to cat allergens and the development of clinical allergy symptoms has been shown in adults and children [7,8,11,12,13]. Clinical manifestations of a cat allergy may include a variety of respiratory but also skin symptoms [13,14,15], and severe asthma due to cat allergy is very common [16].

The molecular characterization of disease-causing allergen molecules has received a strong boost through the application of molecular biological techniques for allergen characterization [17,18]. Today, the molecular structures of many important allergens have been determined, and recombinant allergens resembling the allergen repertoire of complex allergen sources are available, offering possibilities for improved molecular allergy diagnosis and molecular forms of immunotherapy [18,19]. Regarding cats, eight cat allergen molecules have been described and are recorded in the IUIS allergen database [20]. Fel d 1, the major cat allergen, was the first cat allergen molecule for which the primary sequence and three-dimensional structure were determined [21,22]. For a long time, Fel d 1 was considered the sole and most important allergen in cats; accordingly, allergen-specific therapeutic approaches have focused on Fel d 1. The first molecular allergen-specific immunotherapy approaches used non-allergenic peptides from Fel d 1 or hypoallergenic versions of Fel d 1 [23,24,25]. Recently, passive immunization with Fel d 1-specific recombinant human IgG_4_ antibodies was shown to be clinically effective in cat-allergic patients [26]. Regarding diagnostics, IgE sensitization to the major cat allergen, Fel d 1, was even considered as one of the most important marker allergens for severe symptoms of cat allergy in children and for the prediction of respiratory symptoms in adolescence [15,27,28].

However, since then, seven additional cat allergen molecules have been identified, and their IgE recognition frequencies have been determined in cat-allergic patients [29]. Recent studies analyzing IgE cat-allergen-specific IgE levels indicated that high levels of allergen-specific IgE are directed to Fel d 1, Fel d 3, Fel d 4 and Fel d 7 and that these allergens are also frequently recognized by cat-allergic patients [30,31]. However, studies evaluating the allergenic activity of cat allergens, with the exception of Fel d 1, are limited.

In fact, it was recently indicated that not only the frequency of IgE recognition but also the allergenic activity of individual allergen molecules is important for determining the potential clinical relevance of a given allergen molecule; thus, the allergenic activity of individual allergens must be considered in diagnosis and allergen-specific immunotherapy [32]. For example, it was found that two major grass pollen allergens, Phl p 4 and Phl p 13, which were recognized by more than 50% of grass-pollen-allergic patients, induced only mild allergic inflammation when tested via skin-prick testing, whereas others (i.e., Phl p 1, Phl p 2 and Phl p 5) induced strong skin inflammation [33]. The conclusion from the latter study was that Phl p 4 and Phl p 13 contribute only to a low extent to grass-pollen-induced allergic symptoms and do not need to be included in allergen-specific immunotherapy vaccines. Some support for this assumption came from clinical studies performed with a mix of recombinant grass pollen allergens or recombinant hypoallergenic grass-pollen-allergen derivatives comprising Phl p 1, Phl p 2, Phl p 5 and Phl p 6 [34,35].

There are several possibilities to study the allergenic activity of individual allergen molecules: in vivo provocation tests, such as skin tests and nasal, conjunctival and bronchial provocation, and in vitro surrogate tests such as basophil or mast cell activation tests [36]. Skin testing is actually one of the earliest described forms of in vivo provocation [37], but there are more sophisticated methods such as controlled allergen exposure in allergen exposure chambers which also take into account factors such as the mucosal barrier and the concentration of individual allergens in natural allergen sources [38]. However, thus far, there are no exposure-chamber-based approaches for the evaluation of individual allergen molecules, and it is also a technical challenge to expose patients to allergen-containing aerosols. Alternatively, basophil and/or mast cell activation tests can be performed which have been shown to mirror to some extent provocation test results (e.g., skin-prick testing) and in vivo allergenic activity [39,40,41,42]. Of course, there are several limitations of in vitro basophil activation tests, but it is a major advantage that effector cell activation tests allow one to test several allergens in different concentrations on different patients to compare the allergenic potency of the individual allergen molecules under controlled conditions. Our study is the first to compare the allergenic activity for a representative panel of cat allergen molecules with respect to their allergenic activity using basophil activation to provide information regarding the potential clinical relevance of the individual cat allergen molecules.

## 2. Results

### 2.1. Characterization of Cat Allergen Molecules

In a recent study, we characterized the IgE-binding capacity of cat allergen molecules (i.e., Fel d 1, Fel d 2, Fel d 3, Fel d 4, Fel d 6, Fel d 7 and Fel d 8) [30]. Since IgM from cats (i.e., Fel d 6) contains carbohydrate epitopes which fully cross-react with cat IgA (i.e., Fel d 5), Fel d 5 was not included in this previous analysis or in the current study [43]. Here, we evaluated the same panel of cat allergen molecules with respect to their allergenic activity. As a first step, we expressed in *E. coli* and purified recombinant Fel d 1, Fel d 3, Fel d 4, Fel d 7 and Fel d 8 and characterized the purified recombinant allergens using mass spectrometry. Appendix A shows the results obtained from the analysis of the recombinant cat allergens via mass spectrometry. The theoretical molecular masses calculated for Fel d 1, Fel d 3, Fel d 4, Fel d 7 and Fel d 8, including the C-terminal hexa-histidine tags, were 18,942, 11,864, 20,664, 19,302 and 24,766 Da, respectively. These calculated molecular masses were in good agreement with the masses determined via mass spectrometry (i.e., 19,128, 11,695, 20,639, 19,687 and 24,719 Da) (Appendix A).

We then assessed the secondary structure contents of the recombinant allergens via circular dichroism (CD) spectroscopy and found that each of the recombinant allergens showed a secondary structure evidencing that they were folded (Figure 1). Fel d 1 shows a mainly α-helical CD spectrum corresponding to the crystal structure (PDB: 1PUO; [22]) and was therefore well-folded. Fel d 3 exhibits a dominantly β-sheet CD spectrum with a minimum at 215 nm. As there is no experimental structure available, the 3D structure was predicted using a local installation of the ColabFold server [44]. The prediction yielded a structure with a large five-stranded anti-parallel β-sheet and a curved α-helix (prediction with high confidence), and thus indicated that Fel d 3 is correctly folded. Fel d 4 and Fel d 7 both belong to the lipocalin family, and their structures were determined via X-ray crystallography (PDB, 8AMC [45]; PDB, 8EPV [46]). Their structures consist of a nine-stranded β-barrel with an adjacent α-helix. The CD spectra show a dominantly β-sheet fold (with a broad minimum at 215 nm) and are therefore consistent with the experimental structures. Fel d 8 belongs to a family of latherin-like allergens with possible surfactant activity [47,48]. For Fel d 8, the CD spectrum exhibits a minimum at 225 nm and a maximum at 210 nm. The prediction of the structure using the ColabFold server yielded a structure consisting of an elongated four-stranded β-sheet wrapping around two segmented α-helices. The predicted structure exhibits a high structural similarity to members of the BPI fold-containing family A-proteins, although the sequence identity is only around 23% for the best matches [49]. As the CD spectrum is not consistent with the native fold and is also inconsistent with a completely unfolded protein, the protein likely adopts a partially folded structure. Fel d 2 and Fel d 6 were natural cat allergens purified as described.

### 2.2. Characterization of Subjects with IgE Sensitization to Cat Exposure

The subjects included in our study were children (n = 57) aged from 10 to 17 years (mean age: 13.1 years), and there were more male than female subjects (37% females). For the study subjects, only respiratory symptoms upon cat exposure were recorded by the physicians (Table 1 and Appendix A). The majority of children suffered from cat-related rhinitis (i.e., 61.4%), whereas 29.8% suffered from cat-related symptoms of asthma (Table 1). Twenty of the studied children had an IgE sensitization to a cat allergen extract, as determined via quantitative ImmunoCAP measurements, but did not show cat-related respiratory symptoms (Table 1 and Appendix A). Cat-allergen-specific IgE levels varied from 0.19 to 840 kU_A_/L, with a mean level of 53.2 kU_A_/L in all studied children (n = 57) (Table 1). The mean level of cat-allergen-specific IgE in children with respiratory symptoms was 75.94 kU_A_/L, and was thus significantly higher than the mean cat-allergen-specific IgE level in children without respiratory symptoms, which was 11.45 kU_A_/L (Appendix A). All raw data for these results can be deduced from Appendix A. No relevant differences regarding age and gender between children with and without respiratory symptoms were found (Table 1).

In total, 89% of the children had allergic symptoms to pets other than cats (e.g., dogs, horses, other furry animals), and sensitization to pollen (77.3%) and class 1 and 2 food allergens (57.9%) was common (Table 1). Thus, most of the patients also reported symptoms after dog and horse exposure, which may be explained by cross-reactivity between certain animal-derived allergens [28,50,51]. The detailed clinical and immunological characteristics are presented in Table 1 and Appendix A.

### 2.3. Fel d 1, Fel d 4 and Fel d 7 Are the Most Frequently Recognized Cat Allergens and Account for the Majority of Cat-Allergen-Specific IgE

In the next step, we evaluated the frequency of IgE recognition of individual cat allergen molecules in the investigated children (Table 1). The IgE recognition frequencies in the whole study population of subjects with and without cat-related respiratory symptoms were as follows: Fel d 1, 84%; Fel d 2, 26%; Fel d 3, 44%; Fel d 4, 63%; Fel d 6, 26%; Fel d 7, 63%; Fel d 8, 53%. (Table 1). For subjects with and without cat-related respiratory symptoms, the percentages were as follows: Fel d 1, 97% vs. 60%; Fel d 2, 30% vs. 20%; Fel d 3, 49% vs. 33%; Fel d 4, 70% vs. 50%; Fel d 6, 32% vs. 15%; Fel d 7, 68% vs. 55%; Fel d 8, 57% vs. 45%. (Figure 2A). In the whole study population (n = 57), the highest mean specific IgE level was found for Fel d 1 (64.8 kU_A_/L), which was even higher than the mean cat-allergen-extract-specific IgE (i.e., 53.2 kU_A_/L) (Table 1). We found nine children (Appendix A: #2, 8, 9, 17, 37, 41, 43, 46 and 47) who had detectable IgE only to Fel d 1. One of them (Appendix A: #47) was only sensitized to cats and not to other pets, whereas eight were sensitized to other pets, which may be due to a co-sensitization to other pets via genuine allergens occurring only in the other pets. The second and third highest mean IgE levels were found for Fel d 7 (25.15 kU_A_/L) and Fel d 4 (15.4 kU_A_/L) (Table 1). The mean specific IgE levels for the other cat allergen molecules were lower than that of Fel d 1, Fel d 7 and Fel d 4. Six patients had Fel d 1-specific IgE levels of less than 0.1 kU_A_/L (Appendix A: #7, #15, #16, #19, #40 and #48) and showed IgE reactivity only to other cat allergen molecules. In three children (#6, #20 and #53), allergen-extract-specific IgE levels were >0.1 kU_A_/L, but specific IgE levels for the sum of all allergen molecules were below 0.1 kU_A_/L (Appendix A). However, for the latter children, no respiratory symptoms upon cat exposure were recorded. When comparing allergen-specific IgE levels for patients with cat-related respiratory symptoms versus IgE-sensitized subjects without cat-related respiratory symptoms, allergen-specific IgE levels were always much higher for the patients with respiratory allergic symptoms, and this difference was significant for Fel d 1, the cat allergen extract and the sum of Fel d 1–Fel d 8 (Table 1, Figure 2A). Regarding children without respiratory symptoms to cat exposure, the mean IgE level specific for the sum of all tested cat allergen molecules was 22.37 kU_A_/L, and thus was higher than the mean allergen-extract-specific IgE level (i.e., 11.45 kU_A_/L). For the group of children without respiratory symptoms, the mean sum of IgE specific for Fel d 1 + Fel d 4 + Fel d 7 (i.e., 24.6 kU_A_/L) accounted for the majority of the mean IgE level specific for all tested allergen molecules (i.e., 26.3 kU_A_/L).

The mean IgE level specific for the sum of allergen molecules in children with respiratory symptoms (i.e., Fel d 1 + Fel d 2 + Fel d 3 + Fel d 4 + Fel d 6 + Fel d 7 + Fel d 8) was 133.39 kU_A_/L and thus was higher than the mean cat-allergen-extract-specific IgE level (i.e., 75.94 kU_A_/L) in this group. For these patients, the sum of Fel d 1 + Fel d 4 + Fel d 7-specific IgE (i.e., 112.31 kU_A_/L) accounted for the majority of the total sum of allergen-molecule-specific IgE (i.e., a mean of 133.39 kU_A_/L).

The correlation analyses performed in Figure 2B–D showed that cat-allergen-extract-specific IgE levels correlated highly with IgE against Fel d 1 (r = 0.925, *p* < 0.0001) and the sum of Fel d 1–Fel d 8 (r = 0.962; *p* < 0.0001) and with the sum of IgE specific for Fel d 1 + Fel d 4 + Fel d 7 (r = 0.964, *p* < 0.0001). Thus, the correlation of the sum of IgE levels to the Fel d 1–8 allergen molecules and extract (r = 0.962, *p* < 0.0001) was essentially equal to that of Fel d 1, Fel d 4 and Fel d 7. The combination of Fel d 1, Fel d 4 and Fel d 7 allowed us to identify all children with respiratory symptoms to cat exposure via IgE serology (Appendix A).

### 2.4. Allergenic Activity of Cat Allergens

Soon after the discovery of allergen-induced histamine release from leukocytes (i.e., basophils) [52], the basophil activation test was used to assess the allergenic activity of allergens [53,54,55]. Cultured rat basophil leukemia cells expressing the human FcεRI receptor can be loaded with serum IgE from allergic patients to assess the allergenic activity of different allergen molecules under controlled and reproducible conditions [56,57,58,59,60]. Here, we studied the allergenic activity of seven cat allergen molecules (i.e., Fel d 1, Fel d 2, Fel d 3, Fel d 4, Fel d 6, Fel d 7 and Fel d 8) for basophil activation using sera from 17 children with IgE sensitization to cat allergens (Appendix A; Figure 3A,B). All but three (i.e., #12, #16 and #23) of the seventeen children reported respiratory symptoms upon cat exposure. Basophils were loaded with IgE from the patients and exposed to five concentrations of each allergen in tenfold dilutions with the goal of determining the dose-dependency of basophil activation for the individual allergen molecules, which typically is composed of a bell-shaped curve consisting of an increase in mediator release that reaches a plateau and is then followed by a decrease in mediator release [45]. Fel d 1 was the most allergenic molecule because the plateau of maximal release was already reached with the lowest concentration tested (i.e., 0.1 ng/mL; patients #5, #56 and #51) and was reached for all reactive patients at 1 ng/mL (Figure 3A,B; Table 2). Importantly, Fel d 1 induced basophil activation in 88% of the tested patients (Table 2). Fel d 2 showed quite high allergenic activity, reaching the plateau of full activation between ≤0.1 and 10 ng/mL, but only four out of 17 patients showed basophil activation in response to Fel d 2 (Figure 3A,B; Table 2). Fel d 3 was not highly allergenic. It induced full basophil activation between 10 and ≥1000 ng/mL in only 35% of the tested patients (Figure 3A,B; Table 2). Fel d 4 was quite allergenic and caused basophil activation in nine out of the 17 children (i.e., 52.9%), reaching the plateau of maximal release between 0.1 and 100 ng/mL (Figure 3A,B; Table 2). Fel d 6 and Fel d 8 showed very low allergenic activity, triggering full release at a concentration of ≥100 ng/mL in less than 45% of the children. In contrast, Fel d 7 represented a highly allergenic molecule, inducing full basophil activation in almost 65% of the children at a concentration of ≤1 ng/mL (Figure 3A,B; Table 2). High allergen-specific IgE levels were often, but not always, associated with high allergenic activity. For example, in patient #29, the Fel d 2-specific IgE level was approximately tenfold higher than the Fel d 4-specific IgE level in patient #56, but the plateau of full basophil activation was reached in both patients at a concentration of 1 ng/mL. Even if one considers the molar ratios between Fel d 2 and Fel d 4, Fel d 4 was approximately three times as allergenic as Fel d 2 in terms of allergen-specific IgE levels. It has been reported that the ratio of total IgE versus specific IgE may affect the extent of basophil activation [61]. However, when analyzing cat-allergen-induced basophil activation, taking into account total IgE versus allergen-specific IgE levels, we did not find obvious effects of total IgE levels on allergen-specific basophil release. For example, for patients #13 and #23, the ratio of IgE against Fel d 1 to total IgE was 3.29% of total IgE versus 39.59% of total IgE, respectively; however, both patients reached a plateau of mediator release at the same concentration of 1 ng/mL, with release percentages ranging from 50% to 80%. In another patient, #29, the amount of specific IgE to Fel d 1 was 25% of the total level, and specific IgE antibodies to Fel d 7 accounted for only 6.14% of the total IgE, but similar bell-shaped curves for both allergens with similar levels of mediator release were observed.

A sensitivity/specificity analysis of the basophil activation tests was performed with sera from 17 cat-sensitized subjects, 11 non-allergic subjects and nine allergic patients without symptoms of cat allergy. The results in Appendix A show that all cat-sensitized subjects showed basophil activation with at least one of the tested cat allergen molecules. There was no obvious difference regarding the maximal percentage of release for the most allergenic molecule between the patients with symptoms of respiratory cat allergy and those without respiratory symptoms (Appendix A). Only three out of the non-allergic subjects and allergic subjects without cat allergies showed basophil activation with at least one of the tested cat allergen molecules (Appendix A). Thus, in our study population, the sensitivity of basophil testing was 100% and the specificity was 85%, if clinical symptoms to cat exposure were considered the golden standard.

## 3. Discussion

More than 200 million people are allergic to cats, which represent one of the most important indoor allergen sources in the world. Cat-sensitized patients suffer from severe respiratory symptoms such as severe chronic rhinitis and asthma [11,13] The cat represents a complex allergen source comprising several different allergens in addition to the major allergen, Fel d 1 [17,20]. The allergenic activity and clinical relevance of Fel d 1 are well-established [62], but the allergenic activity of the other cat allergens has not been investigated in detail. According to the frequency of IgE recognition and allergen-specific IgE levels, several allergens, in addition to Fel d 1, may contribute to symptoms of cat allergy [29,30]. Our study is the first to evaluate the allergenic activity of known cat allergens (i.e., their ability to induce immediate allergic symptoms). For this purpose, we used two different approaches to study the allergenic activity of the individual cat allergen molecules. First, we investigated the frequency of IgE recognition and allergen-specific IgE reactivity in subjects with and without respiratory symptoms to cat exposure. Fel d 1, Fel d 4 and Fel d 7 were the most frequently recognized cat allergens in the population of patients with respiratory symptoms to cat exposure. The IgE sensitization rates of these allergens ranged from 70% (Fel d 7) to almost 100% (Fel d 1). The quantification of allergen-specific IgE levels for the latter three allergens showed that these allergens accounted for the majority of cat-allergen-specific IgE. The other cat allergens, Fel d 2, Fel d 3, Fel d 6 and Fel d 8, were recognized by a much lower percentage of patients and bound considerably lower levels of allergen-specific IgE.

A comparison of the frequency of allergen recognition and of allergen-specific IgE levels between the group of patients with respiratory symptoms and without respiratory symptoms showed no relevant differences regarding allergen recognition profiles, but allergen-specific IgE levels were much lower in the subjects without respiratory symptoms of cat allergy. One can therefore conclude that allergen-specific IgE levels are associated to some extent with the respiratory symptoms of cat allergy. Neither in this study nor in a recent study [28] did we obtain clear evidence that polysensitization is strongly associated with respiratory symptoms of cat allergy. Regarding the distribution of male and female children in the cat-sensitized children, we noted that there were more males (i.e., 63%) than females (i.e., 37%). In this context, we found one study mentioning that male gender was an intrinsic factor increasing the risk for allergen sensitization [63]. On the other hand, it was reported for a Scandinavian population that girls are exposed to higher levels of cat allergens than boys [64]. However, we think that the number of children investigated in our study is too low to draw conclusions as to whether cat sensitization is linked to a certain gender.

The second and most important approach for the evaluation of the allergenic activity of the individual cat allergens was the testing of their ability to induce basophil activation. In fact, there is good evidence that the basophil activation test reflects clinical sensitivity to allergens very well [39,40,42,65]. For basophil activation, we used a highly reproducible system of loading cultured basophils expressing the human FcεRI with sera from cat-allergic patients and then exposing them to increasing concentrations of the individual allergens. We performed the basophil activation tests with the RBL cell line expressing the human FcεRI because it allows cells to be loaded under standardized conditions with IgE and to test for activation without the presence of interfering IgG antibodies. CD63 activation tests are performed with blood samples which are obtained at different time points; basophils in the blood may exhibit different sensitivities due to the presence of factors which can affect their sensitivity in a non-allergen-specific form (e.g., cytokines), and allergen-specific IgG antibodies may interfere with allergen-induced activation [66]. When testing a concentration of 10 ng/mL of each of the cat allergen molecules, we found that basophil testing with the panel of allergens had 100% sensitivity and 85% specificity to identify cat-sensitized subjects, indicating the usefulness of the basophil test for diagnosis.

Testing different allergen concentrations is very important for determining cellular sensitivity to allergen exposure. In fact, it has been shown in molecular and cellular model systems that the determination of the concentration of an allergen which induces a certain extent of basophil activation is a very useful parameter for the allergenic activity of a particular allergen [67,68]. We therefore calculated the lowest allergen concentration which led to the maximal degranulation of basophils (i.e., the plateau of the bell-shaped curve of mediator release) as a sensitivity indicator for the allergenic potency of a given allergen. The results shown in Table 2 show that Fel d 1, Fel d 4 and Fel d 7 were the most potent allergens, inducing maximal basophil activation at low doses. Fel d 2 also turned out to be highly potent in activating basophils, but was recognized by fewer patients. According to a CD analysis, each of the cat allergens in our study was folded. Only Fel d 8 adopted a partially folded structure. We think that this does not affect the hierarchy of the importance of cat allergens established in our study via IgE binding and basophil activation for two reasons. First of all, Fel d 8 was less frequently recognized by IgE than Fel d 1, Fel d 4 and Fel d 7 in ImmunoCAP testing, which employs large amounts of solid-phase-bound allergen, assuring that sufficient amounts of folded allergen are available to detect all specific IgE. Second, basophil testing showed that Fel d 8 was at least 100-fold less allergenic than Fel d 1, Fel d 4 or Fel d 7. Therefore, even if less than 50% of Fel d 8 was folded, the partial fold by itself cannot account for a 100-fold lower allergenic activity.

We are well aware that the in vitro testing of allergens with respect to basophil activation is only a surrogate for in vivo allergenic activity; nevertheless, our findings seem to indicate that Fel d 1, Fel d 4 and Fel d 7 represent the most clinically relevant allergens which should be considered in molecular approaches for diagnosis and allergen-specific immunotherapy and passive immunization concepts. Our study is thus in agreement with an earlier study measuring allergen-specific antibodies, suggesting that Fel d 1, Fel d 4 and Fel d 7 are the most important cat allergens [29]. It is clear that, in addition to the allergenic activity of individual allergens in basophil activation tests, other factors are also important for the clinical relevance of allergen molecules. Therefore, the limitations of our study are that, in the in vitro test system we used, the influence of blocking IgG antibodies was not considered because the cells were loaded with IgE and IgG was removed. However, this approach is the only one to determine the real allergenic activity of the allergen molecule itself. Furthermore, real-life exposure to certain allergens and the epithelial barrier are other important factors influencing the allergenic activity of different molecules. Finally, the concentrations of the individual allergens may vary in different cats, cat species and in the environment.

However, to the best of our knowledge, our study is the first to provide evidence for the allergenic activity of these individual cat allergens and may serve as a basis for future molecular diagnosis and treatment approaches for cat allergy.

## 4. Materials and Methods

### 4.1. Cat-Allergic Patients’ Sera

Sera from Swedish children with an allergy and/or IgE sensitization to furry animals (n = 57) were obtained from outpatient clinics in the Stockholm area. For these children, only cat-related respiratory symptoms (i.e., rhinitis and/or asthma) were recorded, but symptoms of conjunctivitis and skin allergy were not. Thirty-seven children suffered from cat-related respiratory symptoms, whereas twenty showed only IgE sensitization without respiratory symptoms (Table 1 and Appendix A). For the analysis of the sensitivity and specificity of the basophil activation, testing sera from control groups were included (11 non-allergic donors, NA 1-11, and nine patients with allergies to other allergen sources, but not to cats, NA 12-19 (Appendix A)). The study was approved by the local ethics committees and conducted in accordance with the declaration of Helsinki, and written informed consent was obtained from the parents or legal guardians of all children. A questionnaire was completed for each patient which consisted of questions about demographic data such as age and gender, as well as questions about symptoms (ISAAC questionnaire) that the patients experienced upon contact with a cat. The inclusion criteria for the children were as follows: reported clinical symptoms of either asthma, rhinitis upon exposure to cat and/or IgE antibodies to cat dander extract (e1) greater than 0.1 kU_A_/L, as tested by ImmunoCAP (Thermo Fisher Scientific/Phadia, Uppsala, Sweden). A full description of the clinical and demographic characteristics of children with IgE sensitization to cat exposure and the control groups is shown in Table 1, Appendix A.

### 4.2. Allergen Molecules

rFel d 1 was purified as previously described [62]. Purified natural Fel d 2 was acquired from Sigma-Aldrich (Vienna, Austria), and purified natural Fel d 6 was purchased from Rockland Immunochemicals (Gilbertsville, PA, USA). Recombinant Fel d 3, Fel d 4, Fel d 7 and Fel d 8 were expressed and purified as previously described [30]. The CD spectra of the purified recombinant proteins were determined using a JASCO (Tokyo, Japan) J-810 spectropolarimeter. Measurements were carried out at protein concentrations of 0.1 mg/mL in a rectangular quartz cuvette with a path length of 0.2 cm. Spectra were recorded from 200 to 260 nm with a resolution of 0.5 nm at a scan speed of 50 nm/min and were the result of three scans. Final spectra were corrected by subtracting the baseline spectra obtained with the buffers alone. Results are expressed as the mean residue ellipticities (Θ) at given wavelengths (Figure 1). The allergens’ molecular masses were determined via matrix-assisted laser desorption/ionization time-of-flight mass spectrometry (Bruker, Billerica, MA, USA) (Appendix A) [69].

### 4.3. Allergen-Specific IgE Levels Quantified by ImmunoCAP

The quantitative determination of allergen-specific IgE antibodies to cat dander allergen extract (e1) was performed using ImmunoCAP technology, according to the manufacturer’s instructions, on an ImmunoCAP 100 instrument (Thermo Fisher Scientific/Phadia). Streptavidin ImmunoCAPs (o212 ImmunoCAP, Thermo Fisher Scientific/Phadia) were used to prepare ImmunoCAPs containing Fel d 1, Fel d 2, Fel d 3, Fel d 4, Fel d 6, Fel d 7 and Fel d 8, as described elsewhere [70]. For this purpose, the allergens were dialyzed against a buffer with 0.1 mol/L of NaHCO_3_ and 1 mol/L of NaCl at a pH of 8.0. The allergens were then biotinylated with a five-fold molar excess of biotin (C_26_H_41_N_5_O_7_S, B3295; Sigma-Aldrich, St Louis, MO, USA) at room temperature for two hours. After this, unbound biotin was removed via dialysis against PBS. To determine the optimal allergen amount per cap, for each of the seven allergens, three different amounts of the biotinylated allergen (0.5, 1 and 5 µg) were tested, and the bound IgEs were shown to be comparable. Therefore, 1 µg of biotinylated allergen in 50 µL of PBS was applied to the streptavidin CAPs (#14-532001, Thermo Fisher Scientific, Waltham, MA, USA) [71]. After 30 min of incubation at RT, 150 µL of sera from patients with cat allergies was applied, and IgE testing was performed using an ImmunoCAP Phadia-100 machine, as described by the manufacturer (Thermo Fisher Scientific, Waltham, MA, USA). Allergen-specific IgE levels greater than 0.1 kU_A_/L were considered positive [71].

### 4.4. Rat Basophil Leukemia (RBL) Cells Assay for Testing Allergenic Activity

To test the allergenic activity of the allergens, rat basophil leukemia cells (RBL) expressing the human high-affinity IgE receptor FcεRI (1 × 10^5^/well) were loaded overnight with sera from the cat-sensitized patients, non-allergic donors and patients allergic to other allergen sources at a dilution of 1:10. The cells were washed three times with Tyrode’s buffer (Sigma-Aldrich, Vienna, Austria) and exposed to serial dilutions of the allergens (0.1; 1; 10; 100; 1000 ng/mL) for 1 h. Supernatants were analyzed for β-hexosaminidase activity. Experiments were carried out in triplicates or duplicates with a deviation in the results of less than 5%, and the results are presented as the mean or average percentage of the total β-hexosaminidase released after the addition of 1% TritonX-100. Background values (cells with an allergen without patients’ sera) are shown as a cut-off line.

In order to perform an analysis of the sensitivity and specificity of the basophil activation with cat allergen molecules, sera from 15 cat-sensitized children with respiratory symptoms, two cat-sensitized children without respiratory symptoms, nine allergic patients without cat sensitization (Appendix A) and 11 non-allergic subjects (Appendix A) were tested for basophil activation by exposing IgE-loaded cells to 10 ng/mL of each of the cat allergen molecules. The results are presented in Appendix A as average percentages of release after allergen exposure after the subtraction of the average release + two-fold deviation of the single determinations obtained without patient serum (i.e., the allergen plus the medium).

### 4.5. Statistical Analysis

For statistical analysis, the statistical program GraphPad Prism 6 software (GraphPad Software, La Jolla, CA, USA), was used. The statistical difference between patient groups was calculated using Welch’s *t*-test. Means for the IgE levels in Figure 1 were only calculated for values ≥ 0.1 kU_A_/L. Correlations between groups were determined by calculating Spearman’s correlation coefficient (r), and *p*-values lower than 0.05 were considered statistically significant.

## 5. Patents

The patent application “Vaccine for treating allergies”, application number PCT/CN2022/130414 and reference: P20222370, contains data from this study.

## Figures and Tables

**Figure 1 ijms-24-16729-f001:**
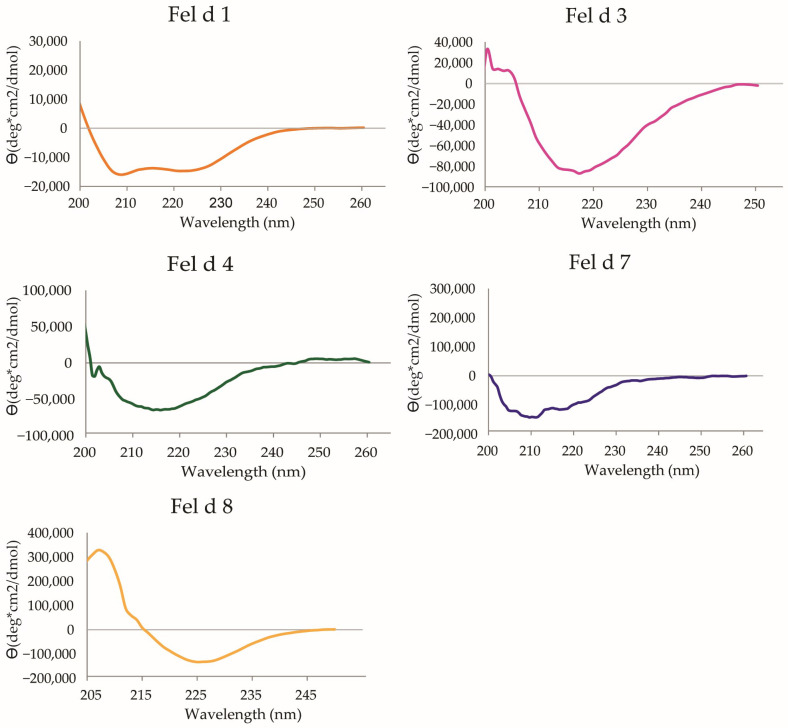
CD spectra of recombinant cat allergens. The mean residue ellipticities (Θ) (y-axes) are shown at given wavelengths (x-axes) for the individual recombinant cat allergens rFel d 1, rFel d 3, rFel d 4, rFel d 7 and rFel d 8.

**Figure 2 ijms-24-16729-f002:**
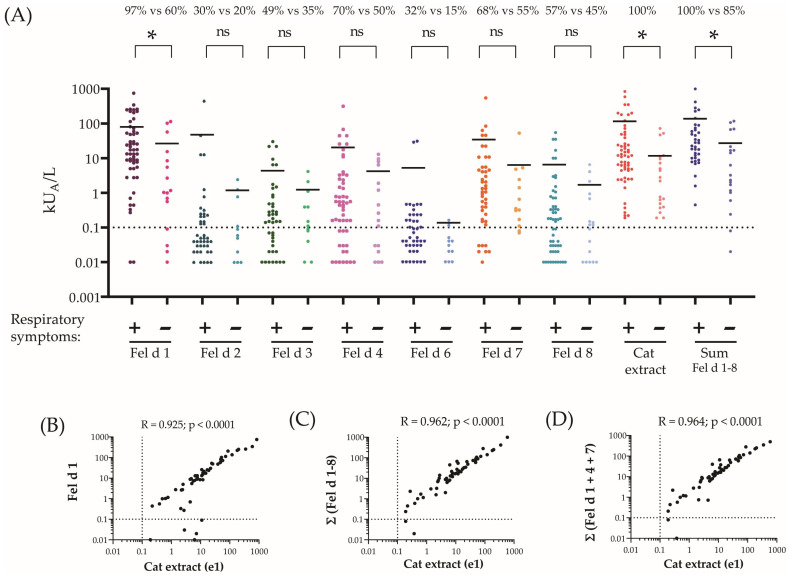
IgE levels to individual recombinant cat allergens and cat hair extract, determined by ImmunoCAP. Specific IgE levels (kU_A_/L, *y*-axis) to cat allergens and a cat extract (*x*-axis), determined for a Swedish population with reactivity to cat extract, are displayed as scatter plots (**A**). The dotted line represents the cut-off value of 0.1 kU_A_/L. Mean IgE levels are presented as horizontal lines. Statistical significance and percentages of IgE-positive sera for the individual allergens are displayed above the plots (significance considered as a *p*-value < 0.05 and indicated by stars; ns, not significant). (**B**) Correlations between IgE to cat allergen extract (x-axes) and Fel d 1 (y-axes), (**C**) cat allergen extract and the sum of IgE against Fel d 1-Fel d 8 and (**D**) between cat-extract-specific IgE and the sum of Fel d 1, Fel d 4 and Fel d 7-specific IgE are displayed as scatterplots. The correlation coefficients and levels of significance (*p*-values) are shown in the upper right corner of the graphs.

**Figure 3 ijms-24-16729-f003:**
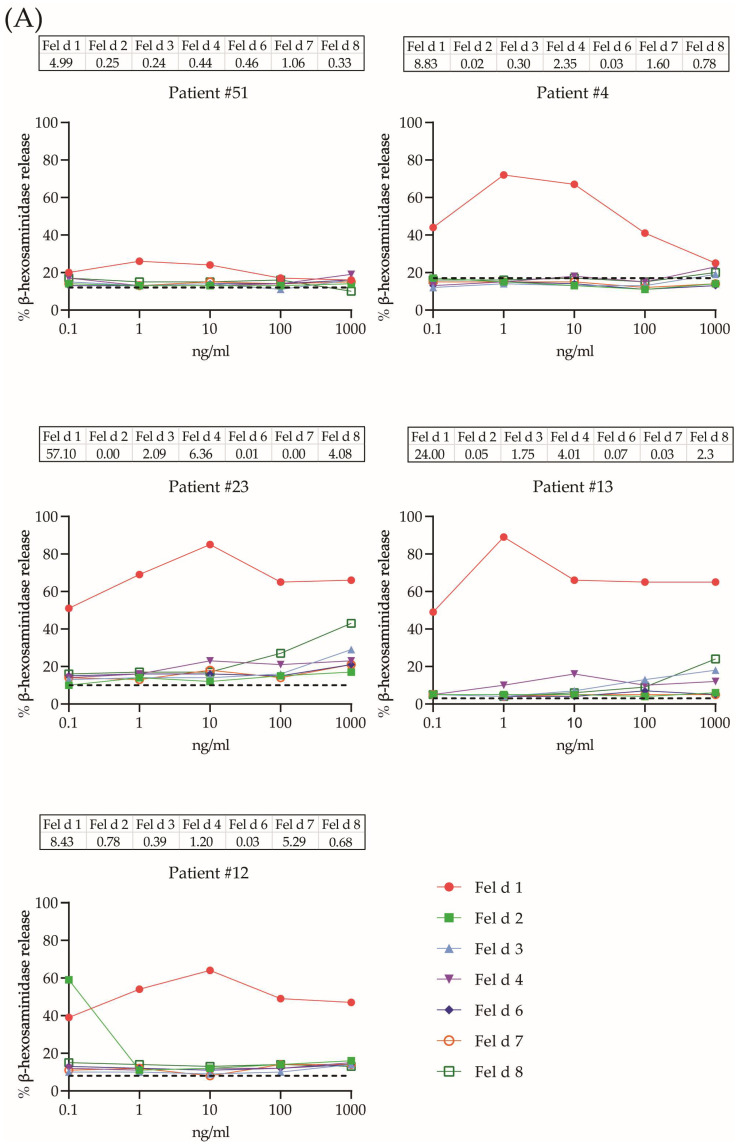
Comparison of the allergenic activity of individual cat allergen molecules. RBL cells were loaded with serum IgE from 17 cat-sensitized children (#4, 5, 7, 10, 12, 13, 16, 23, 24, 26, 27, 29, 34, 36, 51, 57 and 56) and then stimulated with increasing allergen concentrations (0.1 to 1000 ng/mL; x-axes; Fel d 1, Fel d 2, Fel d 3, Fel d 4, Fel d 6, Fel d 7 and Fel d 8, according to color code). Patients were grouped into those showing basophil activation mainly to Fel d 1 exposure (**A**), whereas (**B**) shows patients also showing mediator release to other cat allergens. β-hexosaminidase releases are expressed as percentages of total mediator contents (y-axes). The horizontal cut-off line indicates the percentage of spontaneous β-hexosaminidase release without the addition of allergens. The corresponding IgE levels (kU_A_/L) of the individual cat allergen molecules are shown above the graphs.

**Table 1 ijms-24-16729-t001:** Demographic, serological and clinical characteristics of cat-sensitized patients.

Demographics	All Patients	Respiratory+	Respiratory−	Difference + vs. −
Number of patients	57	37	20	
Age, mean (min–max)	13.1 (10–17)	13.2 (10–17)	12.9 (10–17)	n.s.
Female sex, N (%)	21 (37%)	13 (35%)	8 (40%)	n.a.
**Cat-related clinical symptoms**				
Reported asthma, N (%)	17 (29.8%)	17 (45.9%)	n.a.	n.a.
Reported rhinitis, N (%)	35 (61.4%)	35 (94.6%)	n.a.	n.a.
Reported respiratory symptoms	37 (64.9%)	37 (100%)	n.a.	n.a.
From them, only asthma, N (%)	3 (8%)	3 (8.1%)	n.a.	n.a.
From them, only rhinitis, N (%)	21 (36.8%)	21 (56.7%)	n.a.	n.a.
From them, asthma with rhinitis, N (%)	14 (24.56%)	14 (37.8%)	n.a.	n.a.
**Cat allergy**				
IgE to cat dander (e1), mean kUA/L (min–max)	53.2 (0.19–840)	75.9 (0.2–840)	11.5 (0.19–72)	<0.05
N (%)	57 (100%)	37 (100%)	20 (100%)	
IgE to Fel d 1, mean kUA/L (min–max)	64.8 (0–751)	77.8 (0–751)	25.7 (0–113.5)	<0.05
N (%)	48 (84%)	36 (97%)	12 (60%)	
IgE to Fel d 2, mean kUA/L (min–max)	31.9 (0–446)	46.1 (0–446)	1.15 (0–2.45)	n.s.
N (%)	15 (26%)	11 (30%)	4 (20%)	
IgE to Fel d 3, mean kUA/L (min–max)	3.4 (0–30)	4.2 (0–30)	1.2 (0–4.2)	n.s.
N (%)	25 (44%)	18 (49%)	7 (33%)	
IgE to Fel d 4, mean kUA/L (min–max)	15.4 (0–314)	19.8 (0–314)	4.1 (0–12.9)	n.s.
N (%)	36 (63%)	26 (70%)	10 (50%)	
IgE to Fel d 6, mean kUA/L (min–max)	4.1 (0–30.4)	5.12 (0–30.4)	0.13 (0–0.16)	n.s.
N (%)	15 (26%)	12 (32%)	3 (15%)	
IgE to Fel d 7, mean kUA/L (min–max)	25.15 (0–546)	33.5 (0–546)	6.2 (0–52.5)	n.s.
N (%)	36 (63%)	25 (68%)	11 (55%)	
IgE to Fel d 8, mean kUA/L (min–max)	4.9 (0–54.9)	6.3 (0–54.9)	1.65 (0–6.5)	n.s.
N (%)	30 (53%)	21 (57%)	9 (45%)	
IgE to sum Fel d 1–8, mean kUA/L (min–max)	99.6 (0–1770)	133.4 (0.5–1770)	26.3 (0–116.7)	<0.05
N (%)	54 (95%)	37 (100%)	17 (85%)	
**Other allergy**				
Other pet, N (%)	51 (89.5%)	34 (91.9%)	17 (85%)	
Pollen, N (%)	44 (77.3%)	31 (83.8%)	13 (65%)	
Food (class 1 and class 2 allergens), N (%)	33 (57.9%)	21 (56.8%)	12 (60%)	

Mean allergen-specific IgE levels were only calculated for positive results ≥0.1 kU_A_/L. Abbreviations: Respiratory+ and Respiratory−, patients with or without cat-related respiratory symptoms (asthma or rhinitis); N, numbers; n.a., not applicable; n.s., not significant.

**Table 2 ijms-24-16729-t002:** Summary results of basophil activation by individual cat allergen molecules Fel d 1–Fel d 8.

Allergens	Patients (N = 17)	Plateau of Mediator Release Reached	
	Positive, N/%	Range, ng/mL	Median Conc., ng/mL
Fel d 1	15/88%	≤0.1–1	0.1
Fel d 2	4/23.5%	≤0.1–10	1
Fel d 3	6/35.3%	10–≥1000	100
Fel d 4	9/59.2%	0.1–100	1
Fel d 6	4/23.5%	≥100	≥100
Fel d 7	11/64.7%	≤1	1
Fel d 8	7/41.2%	≥100	≥100

Shown are the ranges of allergen concentrations and the median allergen concentrations reaching the plateau of basophil activation.

## Data Availability

All data are available upon reasonable request.

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
