# Peer review of "Allergenic Activity of Individual Cat Allergen Molecules"

_ijms, 2023, doi:10.3390/ijms242316729_

Round 1
Reviewer 1 Report
Comments and Suggestions for Authors
The study developed by Trifonova and collaborators show that Fel d 1, Fel d 4 and Fel d 7 could be the main responsible allergens for cat allergy in children. This study is consistent, well designed and conducted. I only have a few minor comments to authors:
1. ImmunoCAP and allergenic activity to recombinant Fel d 8 were performed. However this recombinant allergen was characterized as not “completely folded”. Could this affect the results obtained in sIgE determination or β-hexosaminidase release? For example, would sIgE levels be higher if natural Fel d 8 had been used or if the recombinant protein had been obtained correctly folded? It would be needed to add some information about this in the manuscript(for example in discussion section).
2. Please add in Table 1 a new column with statistics between “Respiratory +” and “Respiratory-” in order to compare clinical and demographic data between these groups.
3. Is there any reason why cat allergy is more frequent in female? If there is any reference add it, if not explain why your study population is mostly made up of males.
4. The results from experiments of β-hexosaminidase release are very interesting. In order to have an added value and more potential to this article, is it possible to measure β-hexosaminidase with the serum of all patients (with or without respiratory symptoms) and including some healthy controls? In this way, ROC curves could be calculated and this method could use as possible diagnostic tool to differentiate allergic from healthy, and sensitized from allergic children to cat.
5. From RBL experiments, do you have flow cytometry data? Or only you measured β-hexosaminidase levels in supernatants? It would be interesting to support these results with flow cytometry data on basophil activation, for example by showing CD63 levels.
Comments on the Quality of English LanguageThe english is fine, only a several typos were found.
Author Response
We thank the Reviewer for the kind comments.
- ImmunoCAP and allergenic activity to recombinant Fel d 8 were performed. However this recombinant allergen was characterized as not “completely folded”. Could this affect the results obtained in sIgE determination or β-hexosaminidase release? For example, would sIgE levels be higher if natural Fel d 8 had been used or if the recombinant protein had been obtained correctly folded? It would be needed to add some information about this in the manuscript(for example in discussion section).
Reply: We thank the Reviewer for this thoughtful comment. In fact, we did not mean to say that Fel d 8 is not correctly folded. The Fel d 8 CD spectrum exhibited a minimum at 225 nm and a maximum at 210 nm and the predicted structure exhibited a high structural similarity to members of the BPI fold-containing family Α-proteins. Our conclusion was that the protein likely adopts a partially folded structure. Regarding testing for IgE reactivity by ImmunoCAP where the allergen is present in large excess to IgE it will be still possible to bind all of the specific IgE even is only a fraction of the protein is folded. However, in the ImmunoCAP assays Fel d 8 turned out to be less frequently recognized than Fel d 1, Fel d 4 and Fel d 7 and specific IgE levels were lower than those specific for the aforementioned allergens. Therefore IgE reactivity testing suggests already that Fel d 8 is less important than Fel d 1, Fel d 4 or Fel d 7 and partial fold will not affect specific IgE measurements by ImmunoCAP. According to the basophil testing, Fel d 8 was at least 100-fold less allergenic than Fel d 1, Fel d 4 or Fel d 7 which cannot be explained by the fact that the protein is partially folded because in such a scenario less than 1% of the protein would need to be folded which would not allow to detect partial fold by CD. We have mentioned these considerations in the revised discussion (see lines 371-379).
- Please add in Table 1 a new column with statistics between “Respiratory +” and “Respiratory-” in order to compare clinical and demographic data between these groups.
Reply: We thank the Reviewer for this comment and added the column with the corresponding statistical analysis. There was no significant difference regarding age between the groups and no relevant difference regarding presence of males and females was noted in the two groups.
- Is there any reason why cat allergy is more frequent in female? If there is any reference add it, if not explain why your study population is mostly made up of males.
Reply: According to Table 1 we had more male children with IgE sensitization to cats (i.e., 63% males versus 37% females). We found one study mentioning that male gender was an intrinsic factor increasing the risk for allergen sensitization (Allergic sensitisation in early childhood: Patterns and related factors in PARIS birth cohort. Gabet S, Just J, Couderc R, Seta N, Momas I. Int J Hyg Environ Health. 2016 Nov;219(8):792-800. doi: 10.1016/j.ijheh.2016.09.001. Epub 2016 Sep 13. PMID: 27649627). On the other hand it was reported for a Scandivavian population that girls are exposed to higher levels of cat allergens than boys (Gender differences in indoor allergen exposure and association with current rhinitis. Bertelsen RJ, Instanes C, Granum B, Lødrup Carlsen KC, Hetland G, Carlsen KH, Mowinckel P, Løvik M. Clin Exp Allergy. 2010 Sep;40(9):1388-97. doi: 10.1111/j.1365-2222.2010.03543.x. Epub 2010 Jun 9. PMID: 20545709). We mentioned the fact that there were more boys with cat sensitization in our population and both of the aforementioned studies but think that the number of children investigated in our study is too low to draw conclusions if cat sensitization is linked to a certain gender (see lines 165-166, 336-343).
- The results from experiments of β-hexosaminidase release are very interesting. In order to have an added value and more potential to this article, is it possible to measure β-hexosaminidase with the serum of all patients (with or without respiratory symptoms) and including some healthy controls? In this way, ROC curves could be calculated and this method could use as possible diagnostic tool to differentiate allergic from healthy, and sensitized from allergic children to cat.
Reply: We thank the Reviewer for the comment and performed additional basophil activation experiments with non-allergic subjects (no symptoms of cat allergy), allergic subjects without symptoms of cat allergy (new Table S2) comparing them with cat-sensitized subjects with and without respiratory symptoms. Obtained results were very clear. Results in the newly added Table S3 show that all cat sensitized subjects showed basophil activation with at least one of the tested cat allergen molecules. There was no obvious difference regarding the maximal percentage of release for the most allergenic molecule between the patients with symptoms of respiratory cat allergy and those without (Table S3). Both of the tested cat-sensitized subjects without respiratory symptoms were positive for at least one allergen and high percentages of mediator release were found (see subjects #12 and 16 in the new Table S3). Only 3 out of the non-allergic subjects and allergic subjects without cat allergy showed low mediator release with at least one of the tested cat allergen molecules (Table S3). Thus the sensitivity of basophil testing was 100% and the specificity was 85% in our study population if clinical symptoms to cat were considered as the golden standard. Considering that cat-related allergic symptoms were considered in the determination of sensitivity and specificity it is quite likely that the few reactive subjects had a clinically silent sensitization. This was mentioned in the revised discussion and methods and results have been revised (lines 281-291, 356-358, 407-409, 465-472). In light of the clear results we did not establish a ROC curve.
- From RBL experiments, do you have flow cytometry data? Or only you measured β-hexosaminidase levels in supernatants? It would be interesting to support these results with flow cytometry data on basophil activation, for example by showing CD63 levels.
Reply: We thank the Reviewer for the comment. We have performed the basophil activation tests with the RBL cell line expressing the human FceRI because it allows cells to be loaded under standardized conditions with IgE and to test for activation without the presence of interfering IgG antibodies. CD63 activation tests are performed with blood samples which are obtained at different time points, basophils in the blood may exhibit different sensitivities due to the presence of factors which can affect their sensitivity in a non-allergen-specific form (e.g., cytokines) and allergen-specific IgG antibodies may interfere with allergen-induced activation. Therefore we did not perform a comparison. We have already discussed this topic in the original submission (see lines 347-356, and 387-397).
Reviewer 2 Report
Comments and Suggestions for Authors
Trifonova D et al., presented data regarding molecular characteristics of cat allergens, frequencies of specific IgE to these allergens in children with or without respiratory symptoms when exposed to cat. Authors found that Fel d1, Fel d4 and Fel d7 were the main recognized allergens in sensitized subjects. Levels of specific IgE to cat allergens were higher in patients with than in patients without respiratory symptoms. Data are interesting and clearly presented. I have only a few minor comments.
1. Were children only sensitized to cat, without any other sanitization to pets or other allergens, (they should be 10.5%), monosensitized to Fel d1 or had IgE also to the other cat allergens?
2. Did the authors found any correlation between the ratio “total IgE/allergen specific IgE” of the patients and basophil reactivity induced by the same specific allergen?
3. Abstract lines34-35: the sentence reporting higher levels of IgE to cat allergens in subjects with respiratory symptoms should be moved at the beginning of the Result section, according to description of the results in the text
Author Response
We thank the Reviewer for the comments.
Comments and Suggestions for Authors:
- Were children only sensitized to cat, without any other sanitization to pets or other allergens, (they should be 10.5%), monosensitized to Fel d 1 or had IgE also to the other cat allergens?
Reply: We thank the Reviewer for this useful observation. In fact, we found 9 children (Table S1: #2, 8, 9, 17, 37, 41, 43, 46, 47) who had detectable IgE only to Fel d 1. One of them (Table S1: #47) was only sensitized to cats but not to other pets whereas eight were sensitized to other pets which is not unusual because they may a co-sensitization to other pets via genuine allergens occurring only in the other pets (e.g., Can f 5, etc.). We have mentioned this in the revised result section (lines 190-193).
- Did the authors found any correlation between the ratio “total IgE/allergen specific IgE” of the patients and basophil reactivity induced by the same specific allergen?
Reply: We thank the Reviewer for this smart comment. We determined total IgE levels for our patients and added the results to the revised Table S1. When analysing cat allergen-induced basophil activation taking into account total IgE versus allergen-specific IgE levels we did not find obvious effects of total IgE levels on allergen-specific basophil release. For example, for patients #13 and 23 the ratio of IgE against Fel d 1 to total IgE was 3.29% of total IgE versus 39.59% of total IgE, respectively, however, both patients reached a plateau of mediator release at the same concentration of 1 ng/ml, with release percentages ranging from 50%-80%. In another patient # 29, the amount of specific IgE to Fel d 1 was a 25% of the total level and specific IgE antibodies to Fel d 7 accounted for only 6.14% of total IgE but a similar bell-shaped curve for both allergens with similar levels of mediator release was observed. The examples were added to the revised Results section 2.4 and to the Discussion section (lines 270-280)
- Abstract lines 34-35: the sentence reporting higher levels of IgE to cat allergens in subjects with respiratory symptoms should be moved at the beginning of the Result section, according to description of the results in the text
Reply: We thank the Reviewer for the comment. The sentence was moved to the beginning of the result section (lines 29-30)